# Intramedullary nails versus distal locking plates for fracture of the distal femur: results from the Trial of Acute Femoral Fracture Fixation (TrAFFix) randomised feasibility study and process evaluation

Xavier L Griffin,[1,2] Matthew L Costa,[1,2] Emma Phelps,[1] Nicholas Parsons,[3] Melina Dritsaki,[4] Juul Achten,[1] Elizabeth Tutton,[2,5] Robin Gillmore Lerner,[6] Alwin McGibbon,[7] Janis Baird,[8,9] TraFFix study collaborators

For numbered affiliations see end of article.

**Correspondence to**
Dr Xavier L Griffin;
xavier.griffin@ndorms.ox.ac.uk

## ABSTRACT

**Objectives** This feasibility study and process evaluation assessed the likely success of a definitive trial of intramedullary fixation with locked retrograde nails versus extramedullary fixation with fixed angle plates for fractures of the distal femur.

**Design & setting** A multicentre, parallel, two-arm, randomised controlled feasibility study with an embedded process evaluation was conducted at seven NHS hospitals in England. Treatment was randomly allocated in 1:1 ratio, stratified by centre and chronic cognitive impairment. Participants, but not surgeons or research staff, were blinded to the allocation.

**Participants** Patients 18 years and older with a fracture of the distal femur, who their surgeon believed would benefit from internal fixation, were eligible to take part. Participants were allocated to receive either a retrograde intramedullary nail or an anatomical locking plate.

**Outcomes** The primary outcomes for this feasibility study were the recruitment rate and completion rate of the EQ-5D-5L at 4 months post-randomisation. Baseline characteristics, disability rating index, quality of life scores, measurements of social support and self-efficacy, resource use and radiographic assessments were also collected. The views of patients and staff were collected during interviews.

**Results** Recruitment and data completion were lower than expected. 23 of 82 eligible patients were recruited (nail, 11; plate, 12). The recruitment rate was estimated as 0.42 (95% CI 0.27 to 0.62) participants per centre-month. Data completeness of the EQ-5D-5L at 4 months was 61 per cent (95% CI 43% to 83%). The process evaluation demonstrated that the main barriers to recruitment were variation in treatment pathways across centres, lack of surgeon equipoise and confidence in using both interventions and newly formed research cultures that lacked cohesion.

**Conclusions** A modified trial design, with embedded recruitment support intervention, comparing functional outcome in cognitively intact adults who have sustained a fragility fracture of the distal femur is feasible.

### Strengths and limitations of this study

► Mixed methodology feasibility study including a randomised trial and process evaluation.
► Conducted at seven hospitals, including several major trauma centres.
► Included qualitative interviews with participants, researchers and surgeons.
► The study randomised only a small number of patients.

**Ethics approval** The Wales Research Ethics Committee 5 approved the study (ref: 16/WA/0225).
**Trial registration number** ISRCTN92089567; Pre-results.

## INTRODUCTION
### Background
Distal femur fractures have an incidence of 10 per 100 000 and have a bimodal distribution with age, with approximately 85% sustained by older patients after a fall from a standing height and the remainder sustained by patients after major trauma.[1 2]

There is currently no consensus about the best way to treat these fractures. A recent Cochrane review identified few trials in this area, many using outdated implants or with important methodological limitations and suggested that future patients' functional outcomes could be improved by a well-designed randomised controlled trial comparing modern treatments.[3]

In a multicentre retrospective study to review current practices, we found that the treatments commonly used in the UK are fixation with a locked retrograde nail (nail)

and fixation with an anatomical, angular stable plate (plate).[4] Nails offer minimal disruption of the fracture site, a potential mechanical benefit from a device close to the axis of the femur and stimulated blood supply from reaming of the intramedullary canal.[5] On the other hand fixed-angle plates are specifically designed for osteoporotic bone, and the bone-plate constructs have excellent biomechanical properties.[6] However plates require larger incisions and are more expensive than nails. Limited evidence suggests that there may be a benefit of using nails over plates in terms of patients' quality of life, with small studies identifying differences of 0.09 and 0.1 in EQ-5D-5L-derived utility scores at 1 year.[7 8]

This study of fracture fixation with nails compared with plates aimed to assess the feasibility of a future definitive trial, and perform a process evaluation to understand the generalisability and likely success of a future trial.

## METHODS

### Approval and oversight
The trial protocol is available in the online supplementary material and has been previously published.[9] The conduct of the study was overseen by independent data safety monitoring and trial steering committees.

### Inclusion criteria
Patients aged 18 and over who were admitted to one of seven participating NHS hospitals in England with a fracture of the distal femur, which their treating surgeon believed would benefit from internal fixation, were potentially eligible to take part. Patients with a loose knee or hip arthroplasty requiring revision, or an arthroplasty or pre-existing femoral deformity that precluded nail fixation, were excluded.

All potentially eligible participants for Trial of Acute Femoral Fracture Fixation (TrAFFix) and their carers, as well as all staff involved with the study at participating centres, were eligible to be interviewed for the process evaluation.

### Consent
Where patients had capacity to consent to be involved in the study prior to surgery, this was sought by hospital research teams. For patients lacking capacity to provide informed consent, the hospital research teams sought agreement from an appropriate consultee, either a nominated consultee (eg, treating surgeon who was not part of the research team) or a personal consultee such as the next of kin. Where a consultee gave agreement prior to surgery and the patient regained capacity following surgery, consent for continued participation in the study was sought from the patient.

During the consent process patients and carers indicated their willingness to be approached for interviews for the process evaluation. Consent for interviews was sought at the start of each interview by a researcher from the University of Oxford.

### Randomisation and blinding
Following consent from the patient or agreement from an appropriate consultee, participants were randomised prior to surgery to receive either nail or plate in a 1:1 ratio using an online randomisation system. The trial statistician generated the allocation sequences. Allocations used fixed blocks of size 4 and were stratified by centre to ensure any clustering effect related to centres was distributed across arms, and by the presence of chronic cognitive impairment to ensure this important effect modifier was distributed evenly across groups.

Participants were not informed of their allocation during the trial but were able to request to be informed of their allocation at the end of the study. Surgeons and research staff were not blinded. Radiographs were reviewed by independent assessors, however due to the presence of the implants they were also not blinded.

### Sample size
There was no sample size calculation performed, instead it was estimated that recruiting for 52 centre-months with an average of one patient per centre-month would allow an estimation of the recruitment rate of a future definitive trial with a 95% CI of 0.73 to 1.28.

### Interventions
Potential participants were reviewed in daily trauma meetings and operated on at the next available theatre time. Patients were assessed and received anaesthesia, analgesia and prophylactic antibiotics as per local practices. Preparation, positioning and reduction of the fracture, details of incision and approach and supplementary fixation with wires or screws were left to the discretion of the treating surgeon.

For patients allocated a nail, fixation of the fracture was achieved with a proximally and distally locked nail that spanned the entire diaphysis of the femur. All nails were introduced retrograde through the knee joint. For patients allocated plates, fixation of the fracture was achieved with anatomical distal femoral locking-plate and screws, defined as those in which at least one fixed angle locking screw was placed distal to the fracture.

### Outcome measures
The primary outcomes for this feasibility study were the recruitment rate and completion rate of the EQ-5D-5L[10] at 4 months post-surgery.

### Screening
Participating hospitals were asked to record screening data for all patients admitted with a fracture of the distal femur during the study period. In addition, an orthopaedic surgeon at each participating hospital retrospectively reviewed admissions data and re-screened all admissions during the recruitment period. Similarly, we requested admissions data from the Trauma Audit and Research Network (TARN)[11] in order to monitor the accuracy of screening data from research staff and

orthopaedic trainees. These data included admission date, diagnosis and treatment received.

## Clinical and patient-reported outcomes

Baseline recordings were made of patient characteristics, grip strength[12] and Rockwood frailty scores[13] as predictors of frailty, the level of social support available to patients using the medical outcomes survey social support survey[14] and patient activation using a general self-efficacy questionnaire.[15]

EQ-5D-5L, a generic five-level health utility instrument used to measure health-related quality of life across five domains of mobility, self-care, usual activities, pain and depression was measured at baseline (pre-injury and contemporary) and 6weeks and 4months post-injury. Dementia quality of life scores (DEMQoL, and DEMQoL-proxy when completed by a carer) were recorded for patients with chronic cognitive impairment, along with proxy-reported EQ-5D-5L scores.[16] The disability rating index (DRI),[17] which measures disability from 0 to 100 across 12 items, was recorded at each time point for patients without cognitive impairment.

Radiographs taken at 6weeks post-injury were assessed independently for evidence of loss of fixation, varus or valgus deformity >5° (deformity in the coronal plane), recurvatum/procurvatum >10° (deformity in the sagittal plane) and shortening of the femur >1cm. Related medical complications occurring during the trial period were recorded.

In order to assess the feasibility of a health economical assessment in a larger trial, resource use data were measured using baseline case reporting forms (CRFs) (treatment costs) and a patient completed questionnaire at 4months post-injury. Estimation of unit costs followed recent National Institute for Health and Care Excellence (NICE) guidelines on costing health and social care services[18] using the NHS reference costs[19] and NHS Supply Chain Catalogue[20] and inflated to 2017/18 prices using the NHS hospital & community health services index for health service resources.[21]

## Analysis

Analyses were conducted in R (V.3.3.0, R Foundation for Statistical Computing, Vienna, Austria. https://www.R-project.org/) and the statistical analysis plan is available as online supplementary material. The recruitment rate was estimated using Poisson regression analysis, and the completeness of EQ-5D-5L was calculated as the percentage of randomised participants completing the EQ-5D-5L questionnaire. Given the small study group, treatment effects were not estimated.

Responses from the EQ-5D-5L health classifications were converted into an overall score using a published utility algorithm for the population of the UK.[22]

## Process evaluation

A process evaluation was performed to assess implementation, mechanisms of impact and context of the interventions, in line with Medical Research Council guidelines.[23] In order to inform the development of a definitive trial, our process evaluation also examined the implementation of study processes. The evaluation of implementation included the reach and fidelity of screening and the acceptability of the interventions and study procedures. To understand how the interventions produced change in outcomes, we identified relevant intermediate outcomes that might be associated with the effect of the interventions on the primary outcomes of interest. The context in which the intervention is delivered will influence outcomes. A particular focus in our evaluation of context considered contextual similarities and differences between the participating centres, which might have influenced the delivery of study procedures.

We used a mixed methodology approach using a variety of data sources in the process evaluation including qualitative interviews with patients (n=9), carers (n=2) and staff (n=24), screening logs, national datasets and a 1 day workshop with patients and public representatives. Interviews were based on a topic guide that included participants' experience of taking part in TrAFFix and the experiences of staff recruiting to TrAFFix. Interviews were audio recorded and transcribed verbatim; data were managed using NVivo 10 (QSR International, Warrington, UK). Data were analysed inductively using a thematic analysis. Quantitative data were summarised using standard descriptive statistical techniques. The analyses were integrated using the framework of a process evaluation and summarised in tables.

## Patient & public involvement

A public representative was a member of the trial management committee, and was a co-applicant on the funding application. Patients and public representatives were consulted on the design of patient-facing documents via the UK Musculoskeletal Trauma patient and public involvement mailing list. Patient and public representatives were also invited to a meeting to develop a logical model.

## Important changes to the study design

Patient's age (50 years and older) was initially used as an inclusion criterion as a surrogate for likely fragility fracture. However following review of the screening data, the independent steering committee requested that the minimum inclusion age be lowered to 18 and a mechanistic determination of fragility fracture, using mechanism of injury, be assessed as part of the feasibility study to explore the impact of this modification on recruitment rate.

Instead of utilising the algorithm by Herdman et al[24] to convert EQ-5D-5L to EQ-5D-3L utility as mentioned in the health economics analysis plan, the NICE-approved method of calculating utility values was performed using the algorithm provided by van Hout et al.[22]

## RESULTS
### Recruitment & baseline characteristics
Recruitment opened in October 2016 and closed in August 2017, with a total of 54.8 centre-months of recruitment measured. Twenty three participants were randomised into the study, giving an estimated recruitment rate of 0.42 (95% CI 0.27 to 0.62) participants per centre-month. The flow of participants through the trial is described in figure 1.

Differences in surgical practices were identified across participating hospitals. For example the numbers of patients treated non-operatively varied between 0 and 18 patients (0% to 44% of patients screened at each centre), and the proportion of patients excluded due to surgeon preference varied between 40% and 70%. A breakdown of recruitment and reasons for exclusion for each site is given in online supplementary Table S1.

Of the 173 patients screened, the most common reason that patients were not recruited were surgeon preference for either a nail or plate (23%; 39/173), presence of a pre-existing arthroplasty (19%; 33/173) and a decision to treat the patient non-operatively (18%; 31/173). The number of patients treated non-operatively was higher than anticipated, but confirmed by data from retrospective screening of all admissions, and by data from TARN. The baseline characteristics of randomised participants in each group were well matched and are described in table 1. Following the change of the minimum inclusion age to 18 years old one patient under 50 was randomised, this patient was aged 40 to 45 years, was injured by fall from over 2 m, and had no reported bone health problems.

### Clinical outcomes
Of the 23 participants randomised the EQ-5D-5L questionnaire was completed by 20 (87%, 95% CI 65% to 97%), 15 (65%, 95% CI 43% to 83%) and 14 (61%, 95% CI 39% to 80%) patients at baseline, 6 weeks and 4 months respectively. There were insufficient numbers of patients to draw any meaningful conclusions about treatment effects between groups.

The mean scores across each time point for EQ-5D-5L, DRI and DEMQoL are reported in table 2. Independent assessors at participating sites were asked to review any radiographs taken at 6 week follow-up appointments, the results of assessments are described in table 3, along with a summary of the post-operative complications reported during the 4 month follow-up period.

### Power analysis
We estimated the sample size necessary to power a definitive study using a patient reported outcome such as DRI. Using a minimum important difference of 8 points and

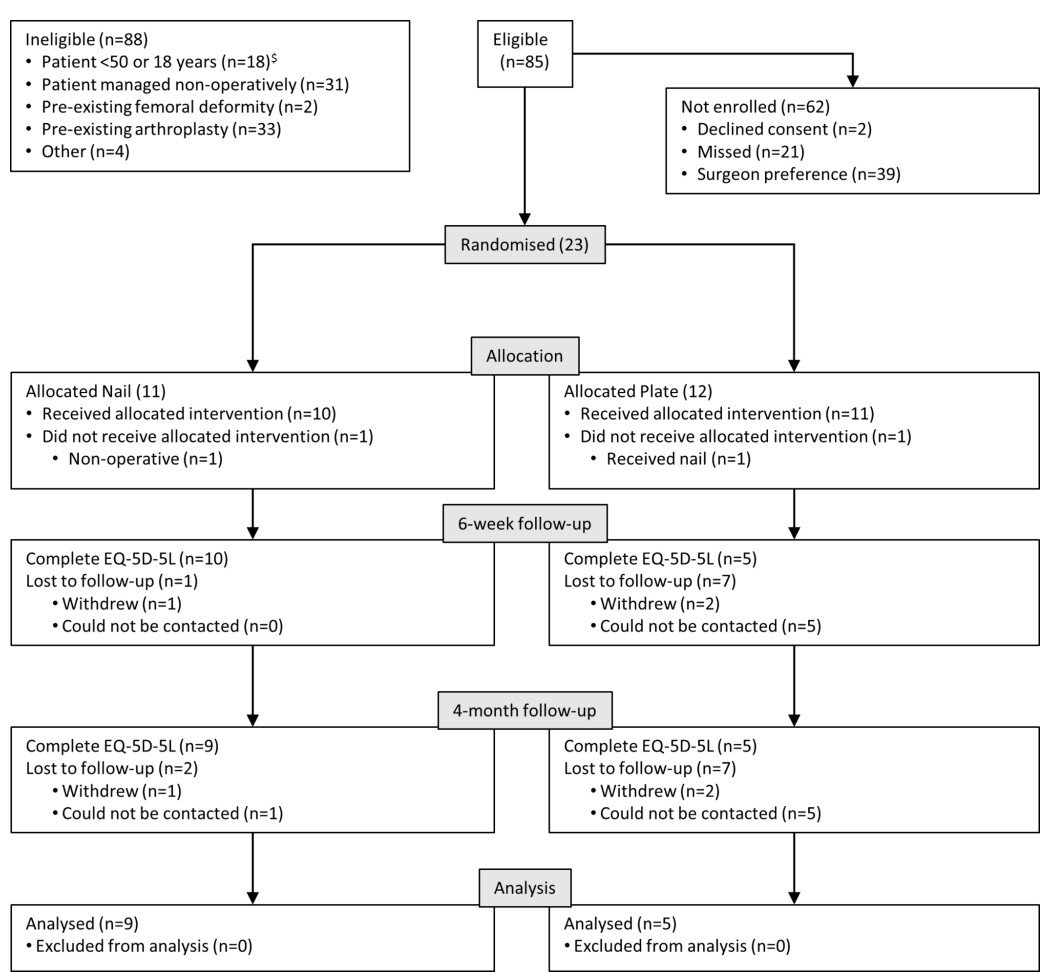

**Figure 1** Summary of recruitment and data collection. $^{\$}$The eligible age was changed from 50 to 18 during the study.

**Table 1** Baseline variables and treatment details

| | Nail (n=11) | Plate (n=12) |
|---|---|---|
| **Baseline variables** | | |
| Age (years); mean (SD; n) | 70.1 (13.6; 10) | 78.7 (14.9; 11) |
| Grip strength (kg); mean (SD; n) | 16.8 (4.6; 4) | 18.7 (1.2; 3) |
| Rockwood frailty score (1 to 9); mean (SD; n) | 3.7 (2.2; 10) | 4.5 (2.1; 12) |
| Pre-Op AMTS | | |
| <7; n (%) | 3 (27.3) | 1 (8.3) |
| 7+; n (%) | 4 (36.4) | 7 (58.3) |
| Post-Op AMTS | | |
| <7; n (%) | 0 (0.0) | 1 (8.3) |
| 7+; n (%) | 2 (18.2) | 3 (25.0) |
| Gender | | |
| Female; n (%) | 7 (63.6) | 9 (75) |
| Male; n (%) | 4 (36.4) | 3 (25) |
| Ethnicity | | |
| Indian; n (%) | 1 (9.1) | 0 (0) |
| White; n (%) | 9 (81.8) | 11 (91.7) |
| Diabetes | | |
| No; n (%) | 5 (45.5) | 9 (75) |
| Yes; n (%) | 5 (45.5) | 2 (16.7) |
| Regular smoker | | |
| No; n (%) | 9 (81.8) | 10 (83.3) |
| Yes; n (%) | 1 (9.1) | 1 (8.3) |
| Living arrangement | | |
| Care home; n (%) | 1 (9.1) | 1 (8.3) |
| Live alone; n (%) | 3 (27.3) | 4 (33.3) |
| Live with relatives; n (%) | 1 (9.1) | 1 (8.3) |
| Live with wife/husband/partner; n (%) | 5 (45.5) | 5 (41.7) |
| Treatment details | | |
| Time from admission to surgery (days); median (IQR) | 2 (1 to 3) | 2 (1 to 3) |
| Mechanism of injury | | |
| Fall from <2 m (%) | 11 (100%) | 12 (100%) |
| Fracture classification* | | |
| A1; n (%) | 7 (63.6) | 5 (41.7) |
| A2; n (%) | 3 (27.3) | 2 (16.7) |
| A3; n (%) | 0 (0) | 1 (8.3) |
| B1; n (%) | 0 (0) | 1 (8.3) |
| C2; n (%) | 1 (9.1) | 1 (8.3) |
| C3; n (%) | 0 (0) | 2 (16.7) |
| Periprosthetic fracture | | |
| No; n (%) | 8 (72.7) | 9 (75) |
| Yes; n (%) | 3 (27.3) | 3 (25) |
| ASA Grade | | |
| One or two; n (%) | 3 (27.3) | 4 (33.3) |
| Three or four; n (%) | 8 (72.8) | 6 (50) |
| Method fixation | | |
| Nail; n (%) | 10 (90.9) | 1 (8.3) |
| Plate; n (%) | 0 (0) | 11 (91.7) |
| Other; n (%) | 1 (9.1) | 0 (0) |
| Grade operating surgeon | | |
| Consultant; n (%) | 7 (63.6) | 8 (66.7) |
| SAS; n (%) | 1 (9.1) | 1 (8.3) |
| ST3+; n (%) | 3 (27.3) | 3 (25) |

Where totals do not sum to column totals then it indicates that there were missing data.
Reported percentages are based on the full population.
*AO/OTA classification of peri-articular fractures.

SD of 21 points defined in previous studies and in line with values reported here,[25] and an attrition rate of 30%, a sample of 210 participants per group (420 in total) would be needed to reject the null hypothesis with probability (power) 0.9 and type I error rate of 5%.

**Process evaluation**

A summary of the complete findings of the process evaluation is presented in the online supplementary table S2.

**Table 2** Patient reported outcome measures

| | Nail (n=11) | Plate (n=12) |
|---|---|---|
| **Pre-injury: mean (SD; n)** | | |
| EQ-5D-5L | 0.59 (0.29; 10) | 0.56 (0.22; 10) |
| DRI | 45.4 (36.7; 9) | 67.5 (10.1; 10) |
| DEMQoL proxy | 87 (0; 1) | – |
| DEMQoL self | – | 78.2 (0; 1) |
| **Post-injury: mean (SD; n)** | | |
| EQ-5D-5L | −0.05 (0.28; 10) | −0.04 (0.16; 10) |
| DRI | 86.5 (15.5; 9) | 92.5 (5.3; 10) |
| DEMQoL proxy | 97 (0; 1) | – |
| DEMQoL self | – | 84 (0, 1) |
| **6 weeks: mean (SD; n)** | | |
| EQ-5D-5L | 0.35 (0.30; 10) | 0.05 (0.16; 5) |
| DRI | 78.0 (16.9; 9) | 87.3 (3.1; 5) |
| DEMQoL proxy | 87 (0; 1) | – |
| DEMQoL self | – | – |
| **4 months: mean (SD; n)** | | |
| EQ-5D-5L | 0.38 (0.36; 9) | 0.37 (0.41; 5) |
| DRI | 60.9 (23.1; 8) | 82.8 (2.9; 4) |
| DEMQoL proxy | 89 (0; 1) | – |
| DEMQoL self | – | – |

*Indicates no data were available. DEMQoL, Dementia quality of life scores; DRI, disability rating index.

**Table 3** Summary of 6 week postoperative radiograph review and reported complications reported up to 4 months

| | Nail (n=11) | Plate (n=12) |
|---|---|---|
| Six week radiographic assessment: n (%) | | |
| Loss of fixation | 1 (9.1) | 1 (8.3) |
| Varus/valgus >5° | 10 (90.9) | 5 (41.7) |
| Recurvatum >10° | 1 (9.1) | 0 (0) |
| Procurvatum >10° | 1 (9.1) | 0 (0) |
| Shortening >1 cm | 2 (18.2) | 0 (0) |
| Postoperative complications: n (%) | Nail | Plate |
| Wound infection | 0 (0) | 0 (0) |
| Venous thromboembolism | 0 (0) | 0 (0) |
| Pneumonia | 1 (9.1) | 0 (0) |
| Urinary tract infection | 1 (9.1) | 0 (0) |
| Cerebrovascular accident | 0 (0) | 0 (0) |
| Myocardial infarction | 1 (9.1) | 0 (0) |
| Blood transfusion | 2 (18.2) | 2 (16.7) |
| Malunion | 0 (0) | 0 (0) |
| Failure of fixation | 0 (0) | 1 (8.3) |

## Implementation

### Reach of screening

During the 10 months recruitment period, 91 patients were screened and recorded on screening logs, usually by a research associate (RA). Of these, 54 were eligible to participate in the study. Retrospective review of screening by orthopaedic surgeons at each participating centre found a further 82 unscreened patients with distal femur fractures including 31 eligible patients who were not included on the screening logs. The number of patients added after rescreening varied by centre with between 0 and 34 patients added. At interview, staff emphasised that there were few patients eligible for the study, suggesting that they were unaware that potentially eligible patients were being missed.

'With the TrAFFix study, well we just never had the numbers coming through and I think the only two that we actually missed were during weekends and I think one was when I was on annual leave' Staff (RA) 19

The differing levels of experience of the research teams and research culture, evident in the approaches to screening, recruitment and data collection, within the centres may explain differences in the number of patients missed from screening logs. In some centres, not all surgeons within the team screened and identified eligible patients, while in other centres patients deemed ineligible might not have been recorded in the screening logs.

### Fidelity of screening

Interviews with research staff revealed that several centres relied on research teams to screen patients as the clinical team tended not to notify the research team of eligible patients. Screening was facilitated by the presence of the research team in the daily trauma meeting, as they were able to prompt the clinical teams to consider whether patients were eligible. The attendance of the research team in this meeting varied between the centres due to staffing and other commitments. Additionally, some centres found it difficult for the clinical team to keep the study in mind as potentially eligible patients were infrequent.

### Acceptability of the interventions

At interview, some staff described an unwillingness from surgeons to randomise patients. They felt that some surgeons lacked equipoise and believed that only one method of fixation was appropriate for any given fracture.

'you would never do the other thing for this fracture' Staff (Surgeon) 8

However, some staff reported that surgeons' preferences for one of the two interventions were based on their surgical skill and experience.

'Some surgeons, they only know how to do a plate and what you will see is plate plate plate plate so it comes down to also what you are good at sometimes and you don't want something else.' Staff (Surgeon) 16

In contrast some surgeons were able to accept community equipoise (uncertainty within the expert medical community),[26] accepting their less preferred intervention if required.

### Acceptability of the study procedures

#### Consent

Participants and carers were rarely able to describe the study in their own words but they tended to recall or recognise the interventions.

'Yes, because one is on the outside and one is a rod through the middle isn't it' Participant 4

Several participants described struggling to engage with information about the study around the time of their surgery explaining that they were not 'in a fit state' (participant 5) to ask questions or were 'trying to so hard to be normal' (participant 3).

Interviews with staff showed that they described the study to patients in simple terms. Randomisation, for example was explained using phrases such as '50/50' or 'computer decides.' When explaining the study, staff also emphasised that both treatments are routinely used and that their surgeon was happy for them to receive either intervention, as they believed these were important to patients.

 Griffin XL, et al. BMJ Open 2019;9:e026810. doi:10.1136/bmjopen-2018-026810

'They're going to have surgery anyway and then like I said if you say we don't know which the better one is and the surgeon is happy, I think is the key point is that the surgeon is happy for them to be part of the study' Staff (RA) 1

The majority of staff found involving relatives in the discussion about the study to be helpful as it enabled patients to be supported in their decision-making. In contrast, a minority of staff felt that it was easier to consent patients to studies when their relatives were not involved as they could be protective and involving them could lead to more potential participants declining.

'Sometimes you can find it easier if you approach them to try and time it around their visiting hours and so they've got a relative with them and so you can kind of talk to them as a family and a lot more patients feel more comfortable with that' Staff (RA) 7

Two of seven patients who were entered into the study under nominated consultee agreement withdrew when they were approached to consent to continue in the study after surgery. One of these seven patients participated in an interview. She acknowledged that she '*wasn't sufficiently with it to give an opinion*' (participant 1) prior to surgery.

### Randomisation

The majority of staff found randomisation was acceptable to patients, explaining that it was considered acceptable because of the explanation that the surgeon thought both treatments were appropriate. Two members of staff, in contrast, found patients disliked randomisation and wanted their surgeons to choose the treatment that was most appropriate for them.

When randomisation was described to participants during their interview, they accepted it or seemed indifferent towards it. Several participants, however, demonstrated therapeutic misconception: '*when a research subject fails to appreciate the distinction between the imperatives of clinical research and of ordinary treatment, and therefore inaccurately attributes therapeutic intent to research procedures*'.[27] They believed that they or their relative would receive the most appropriate intervention for them.

'I think I prefer the one they done. I mean I didn't have a say in it, they decided what they thought was best, you know but I preferred the one they done definitely.' Participant 6

### Blinding

A minority of participants were told of their treatment allocation following surgery or when they returned for follow-up appointments. Some staff felt that blinding participants to their treatment allocation would be difficult as patients could see their allocation in the letter sent to their general practitioner or X-rays, or they may be told by another member of staff. Some staff wanted to tell patients who asked which intervention they were allocated as one explained, '*I wouldn't be particularly comfortable, on*

*a personal level… I would want to tell them what they have had*' (surgeon 8).

### Case reporting forms

Research staff found that completing CRFs with the patient group could be challenging, as most patients were frail and often had long-term medical problems. They felt the questionnaires were lengthy and tiring for patients who often needed support completing them. They tended to ask the questions as part of a conversation to pull out the information for the questionnaire as they found some patients were unable to express their experience using the responses in the scales. Several RAs felt that where patients were wary of participating in studies this was usually due to the burden of follow-up rather than a dislike of randomisation or preference for one of the interventions.

'Some of the questions you have to try to put it into different words for them to understand, some of the questions don't make any sense to them' Staff (RA) 2

### Mechanisms of impact

We identified intermediate outcomes that might be associated with the effect of the interventions on the primary outcomes of interest. Associations between the factors identified here, the interventions and the outcome of interest will be assessed in a definitive trial. Factors that might influence patients' recovery after a distal femur fracture were identified by patients and public representatives and from data collected from interviews with participants and staff. Factors fell into three groups: existing factors relating to the characteristics of patients including their age, cognition and psychological factors such as self-efficacy; injury and treatment factors including the injury itself and factors relating to the surgeon and centre in which the treatment was delivered; post-discharge factors including rehabilitation provided, living arrangements and availability of support at home both from social care and from family. These factors are presented in online supplementary table S3.

### Context

Contextual differences between the participating centres may have led to differences in how the study was implemented. Surgeons' working in departments where several trials were running may have been more engaged in research and more inclined to participate.

'However, that's changing and we've got a lot more of the consultants on board now and a lot more involved in the trial… there's consultants that are running trials in hand orthopaedics, the more elective side of things but also do the trauma lists and so are getting more involved with what's going on in research and therefore they're more happy to be involved in it and identifying the patients and things like that.' Staff (RA) 4

Some staff felt that differences in the size of the research team, their workload and working hours could influence how the study was implemented. For example, the presence of RAs in the trauma meeting could facilitate screening but required RAs to start work before eight, which was not possible in every centre.

'I'm keen to see what is going to happen when we do get the other research nurses going to the trauma meeting because I feel that that's going to make a big difference and I think our recruitment will increase quite a lot but I might be completely wrong in thinking that.' Staff (RA) 4

'I think if we didn't have enough studies running to have a fully funded research team then it would be tricky.' Staff (surgeon) 20

Meanwhile, some centres relied on one or part-time RAs. This can result in patients being missed either when the RA was away or because they were struggling with their workload.

'There was nobody in (to recruit) this Friday just gone and it can happen because I've got to be everywhere.' Staff (RA) 16

## DISCUSSION

We have reported a randomised feasibility study, and to our knowledge the first with an embedded process evaluation, to plan a future trial of the treatment of patients with a fracture of the distal femur. We found the number of eligible patients screened closely matched the original estimate of 1.5 patients per centre-month. The recruitment rate into the trial, however, fell substantially short of this. The process evaluation identified very significant surgeon-related barriers to recruitment into the study. Research cultures were found to vary widely between centres; a positive culture was a strong predictor of improved recruitment performance.

This study has challenged many of the assumptions which underpinned the development of the trial protocol. The recruited population had a much lower prevalence of cognitive impairment, less frailty and was physiologically fitter than the previously characterised incident fracture population.[4] Surgeons were actively selecting fitter patients for operative treatment in a way that we had not anticipated. The eligible population is therefore somewhat different than we originally expected and hence the design for the definitive trial can be modified in important ways.

It is clear that research staff can and do miss potentially eligible patients if the working practices at hospitals do not facilitate full integration of the research staff into the clinical team. Although the specifics vary between hospitals, crucial decision points exist within the clinical process, such as daily trauma meetings, into which research staff must be integrated in order to be able to engage clinicians in research studies.

Surgeons reported a lack of their confidence in the management of this relatively uncommon fracture, the operative treatment of which is perceived to be technically demanding. The combination of this common anxiety and some individuals' lack of equipoise, posed a very significant barrier to recruitment into this study. Similar challenges have been previously identified in other surgical trials.[28 29] Interventions that have included training on community equipoise have been successful when targeted at multidisciplinary teams and surgeons, improving both clinicians' confidence and potentially recruitment rates,[30 31] though definitive evidence is lacking.[32] We propose that a modified protocol that includes an intervention to promote awareness and understanding of the existing community equipoise will be able to overcome these surgeon-related barriers to recruitment.

Our findings demonstrate that the trial design tested in this feasibility study is unlikely to be successful on a larger scale. However, the recruitment rate of 0.42 participants per centre-month is in line with that reported by the FixDT trial, which tested similar interventions for treating fractures of the tibia and delivered to target and budget.[25]

We propose a modified definitive trial with a further internal pilot to confirm recruitment rates; this will be based on the findings of this feasibility study, with an integrated recruitment intervention to support surgeons in their decision-making to recruit participants.

A future study may choose to exclude the small number of patients with chronic cognitive impairment to simplify trial procedures for research staff or reduce the number and length of questionnaires to reduce the burden on participants and improve retention rates. Similarly a future study may consider a trial design that accounts for surgeons' treatment preferences during the allocation of treatment to improve surgeon participation and willingness to randomise.

We conclude that a trial comparing functional outcome in cognitively intact patients aged 18 years or older who have sustained a fragility fracture of the distal femur, defined as a fall from standing height, treated with a nail or plate, is feasible with appropriate mitigation strategies incorporated to avoid failure.

**Author affiliations**
[1]Oxford Trauma, University of Oxford Nuffield Department of Orthopaedics Rheumatology and Musculoskeletal Sciences, Oxford, UK
[2]Oxford University Hospitals NHS Foundation Trust, Oxford, UK
[3]Statistics and Epidemiology Unit, University of Warwick, Coventry, UK
[4]Oxford Clinical Trials Research Unit, University of Oxford Nuffield Department of Orthopaedics Rheumatology and Musculoskeletal Sciences, Oxford, UK
[5]Warwick Medical School, University of Warwick, Coventry, UK
[6]Kadoorie Centre, University of Oxford Nuffield Department of Orthopaedics Rheumatology and Musculoskeletal Sciences, Oxford, UK
[7]Patient and Public Representative, Coventry, UK
[8]MRC Lifecourse Epidemiology Unit, University of Southampton, Southampton, UK
[9]NIHR Southampton Biomedical Research Centre, University Hospital Southampton NHS Foundation Trust, Southampton, UK

**Collaborators** Mr David Noyes, John Radcliffe Hospital, Oxford University Hospitals NHS Foundation Trust. Professor Peter Giannoudis, Leeds General Infirmary, The Leeds Teaching Hospitals NHS Trust. Professor Ben Ollivere, Queen's Medical Centre, Nottingham University Hospitals NHS Trust. Ms Charlotte Lewis, Queen Alexandra Hospital, Portsmouth Hospitals NHS Trust. Mr Haroon Majeed, Mr Damian McClelland, Royal Stoke Hospital, University Hospitals of North Midlands NHS Trust. Mr Ashwin Kulkarni, Leicester Royal Infirmary, University Hospitals of Leicester NHS Trust.

**Contributors** XLG (Associate Professor of Trauma Surgery) contributed to the conception, design, conduct and reporting of the study. MLC (Professor of Orthopaedic Trauma Surgery) contributed to the conception, design and reporting of the study. EP (Postdoctoral Research Associate in Mixed Methods Research) contributed to the design, conduct and analysis of the process evaluation and the reporting of the study. NP (Associate Professor of Medical Statistics, Statistics and Epidemiology) contributed to the design, conduct, analysis and reporting of the study. MD (Senior Health Economist) contributed to the design and conduct of the study. JA (Scientific Officer Oxford Trauma) contributed to the conception, design, conduct and reporting of the study. ET (Senior Research Fellow) contributed to the design, conduct and analysis of the process evaluation and the reporting of the study. RL (Clinical Trial Manager) contributed to the conduct and reporting of the study. AMG (Patient & Public Representative) contributed to the design, conduct and reporting of the study. JB (Professor of Public Health and Epidemiology) contributed to the conception, design, conduct, reporting and analysis of the study.

**Funding** This project was funded by the Health Technology Assessment programme (HTA) (project number 15/59/22) and will be published in full in the NIHR HTA Journal. Further information available at: https://www.journalslibrary.nihr.ac.uk/programmes/hta/155922/#/. This work was also supported by NIHR Oxford Biomedical Research Centre.

**Competing interests** Matthew Costa is a member of the UK NIHR HTA General Board. Janis Baird received funding from Danone Nutrica Early Life Nutrition for a specific research study which aims to improve the nutrition and Vitamin D status of pregnant women and is collaborating with Iceland Foods Ltd to evaluate the impact fruit and vegetable availability on diet.

**Patient consent for publication** Not required.

**Provenance and peer review** Not commissioned; externally peer reviewed.

**Data sharing statement** All data requests should be submitted to the corresponding author for consideration. Please note exclusive use will be retained until the publication of major outputs. Access to anonymised data may be granted following review.

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
