## [Reviewer comments · BMJ Open]

ARTICLE DETAILS

TITLE (PROVISIONAL)	Intramedullary nails vs distal locking plates for fracture of the distal femur: results from the Trial of Acute Femoral Fracture Fixation (TrAFFix) randomised feasibility study and process evaluation
AUTHORS	Griffin, Xavier; Costa, Matthew; Phelps, Emma; Parsons, Nicholas; Dritsaki, Melina; Achten, Juul; Tutton, Elizabeth; Lerner, Robin; McGibbon, Alwin; Baird, Janis

VERSION 1 - REVIEW

REVIEWER	Hannu T Aro Turku University Hospital and University of Turku
REVIEW RETURNED	23-Nov-2018

GENERAL COMMENTS	This is a well-written, important report about challenges to execute a randomized clinical trial (RCT) in orthopaedic trauma patients. Correctly, the investigators have performed a limited feasibility study with an embedded process evaluation. A randomized comparison of the two surgical techniques in fracture fixation of the distal femur seems to be justifiable. A recent register analysis (Hoskins et al. Bone Joint J 2016), co-authored by the first author, showed that intramedullary nailing may be superior treatment compared with anatomical locking plating. However, the analysis of the retrospective data was biased because surgeons chose to treat more severe intra-articular fractures with locking plates. Unfortunately, the current feasibility study demonstrates that a RCT approach does not solve this bias but brings even new challenges. A reader may disagree about the conclusions of this study. Under the clinical research conditions of the investigators, a RCT on the operative treatment of this fracture may not be feasible. Indeed, a RCT is not an easy way to get definite answers in orthopaedic trauma surgery. As stated by the investigators, the fractures of the distal femur are relatively rare and, at the same time, their operative treatment is technically demanding. Therefore, it might be unfair to force even experienced surgeons to use a method they do not use routinely. A trial should not study the impact of a learning curve. It is notable that in this study the portion of patients excluded due to surgeon preference varies between 40% and 70%. So, this was not a problem related to single surgeons or centres but a uniform issue among the clinical investigators of the recruited centres. An unorthodox approach, using a parallel non-randomized prospective head-to-head comparison of the two surgical methods (based on the personal or
---

	center-based preference), might be the only choice and even bring even more relevant data. Specific comments:  1. It is mandatory that the manuscript includes a separate section for power analysis of the proposed final trial. The investigators have now retrospective data (n=297)(Hoskins et al. Bone Joint J 2016) and the current data (n=23). The data should allow the estimation of the sample size for the selected primary outcome. Looking the retrospective study, the final study may need about 150 participants per group. 2. Looking the data, it seems to be challenging to run a final trial within an acceptable time period. Taking the estimation rate of 0.42 participants per centre-month and the need of about estimated 300 participants, it will take 8.5 years to execute the final trial in seven centres. This is not acceptable. A long recruitment period will increase the risk of uncontrollable confounding variable, as seen in previous trials. 3. In Discussion, the authors propose a modified definite trial with an integrated recruitment intervention to support the decision-process of surgeons. However, no clear-cut answers are given how to prevent the high rate of exclusion due to surgeon preference of the fixation modality. 4. Based on the protocol and for a good reason, patients with pre-existing arthroplasty were ineligible. However, 5 subjects were included in this study although they had a periprosthetic fracture. 5. As a related issue, if the 5 subjects with periprosthetic fractures were excluded, the feasibility study was able to recruit only 18 subjects within the 10-month recruitment period. The study was sponsored by the National Institute of Health Research Technology Assessment. It is reasonable to evaluate the study cost-effectiveness especially because there are many other important competing topics in the field of orthopaedic trauma surgery. What was the plain cost per a recruited subject? 6. The issue of practical problems in the execution of RCT of fracture patient is not new (Csimma & Swiontkowski J Bone Joint Surg Am 2005;87: 218-222). It is important to avoid situations where a trial must be prematurely stopped due to recruitment problems. The authors should consider to cite examples from the literature (Bhandari et al. Clin Orthop Relat Res 2016; 474: 1234-1244) (Inngul et al. Bone Joint J 2015;97-B: 1475-1480). 7. The controversial issue how to fix the intra-articular fractures of the distal femur should be discussed.
--	--

REVIEWER	MEHMET SALIH SOYLEMEZ UMRANIYE TRAINING AND RESEARCH HOSPITAL DEPARTMENT OF ORTHOPAEDICS AND TRAUMATOLOGY ISTANBUL/TURKEY
REVIEW RETURNED	23-Dec-2018

GENERAL COMMENTS	This is a well planned and written feasibility study with relatively small number of individuals. Authors in their previous paper(1) had defined the “fragility fracture of the distal femur” as a fracture of distal femur after a low energy trauma in patients older tan 50 years-old. Authors in the same paper had removed the age barrier to decrease the number of
---

	patients excluded from the study. However in the present paper patients over 18 years old have been included, and only one patient have reported to be under age of 50. If so, What was the trauma of this particular patient? What was the age? Sustaining a fracture after a low energy trauma doesn't means that it should be a "fragility fracture" Does this patient had a cyst in the bone? or what. This must be clarified and definition of "fragility fracture" fort he age between 18 and 50 must be detailed . 1. Griffin XL, Costa ML, Achten J, et al. Trial of Acute Femoral Fracture Fixation (TrAFFix): study protocol for a randomised controlled feasibility trial. Trials 2017;18(1):538. doi: 10.1186/s13063-017-2265-0
--	--

REVIEWER	David Colquhoun UCL UK
REVIEW RETURNED	30-Jan-2019

GENERAL COMMENTS	It is very good to see proper trials being done to test surgical procedures. There should be more of them. The results are quite depressing. The fact that the most common reason for failing to recruit was surgeon preference. Right from the start of RCTs they have been opposed by people who thought they knew the answer. The problem, of course, was that different people "knew" different answers, It would be interesting to know the extent to which surgeon preferences were based on prejudice and to what extent they are based on skills. If a surgeon is practiced in only one of the two methods, it would not be a fair comparison if he were forced to do the other. Nonetheless "unwillingness from surgeons to randomise patients" suggests that many surgeons still fail to realise the importance of evidence. The large dropout rate for follow-ups surely suggests that the people on the ground needed better training. The conclusion that cognitively-impaired patients should be excluded in future work sounds eminently sensible to me. It can only add noise to try to compare the different sorts of outcome measures that are needed for those with and without cognitive impairment. I find it a bit concerning that one of the team has "received funding from Danone Nutrica Early Life Nutrition", given the number of times this company has been condemned for dishonest advertising. This connection might well harm the credibility of the results. I'm also a but concerned about "The principles are not intended to curtail exploratory analysis (for example, to decide cut-points for categorisation of continuous variables), " Cut-points to categorises continuous variables waste information and can be misleading. Why categorise at all? Surely the optimum analysis would be to perform randomisation tests using the original numbers.
--

	The statement "primary outcome and key secondary outcomes will be analysed, following the analyses detailed in this SAP, by a statistician independent of the trial using different statistical software (if possible)." is excellent. It should be done more often. But why not mention randomisation tests? There is software available to do this, and, in any case, it's easy to write the necessary software in R. One thing I can't agree with is "significance levels used will be set at the conventional 5% level.". The one thing that essentially all statisticians now agree on is that the term "statistically-significant" should be dropped. The myth that $p < 0.05$ is good evidence has surely been superseded now. My solution would be to give, as well as p value and CI, a measure of the (minimum) false positive risk. Blinding is obviously crucial, so patients should not be told which procedure they got until after the end of the follow-up period. But perhaps the biggest problem is the skills of surgeons. I don't know how many surgeons feel equally confident to use both methods. I'd guess not many do. If this is so, it would be good if patients could be allocated randomly to method A or B but then operated on only by surgeons who are practised in the allocated methods. I don't know how feasible this would be in practice, but it doesn't make much sense to compare treatments A and B, if A is given by someone who has done A many times, but B is done by someone who has never done B before.
--	---

VERSION 1 – AUTHOR RESPONSE

We thank reviewer 1 (Hannu T Aro) for their comments, and agree that traditionally designed RCTs are difficult in this field. Future studies will need to employ and assess strategies and designs that help collect robust data in this area. We have address their comments in order.

In response to the comment (1) that a section of power analysis of a definitive trial should be added, we have now included the following section:

"We estimated the sample size necessary to power a definitive study using a patient reported outcome such as DRI. Using a minimum important difference of 8 points and standard deviation of 21 points defined in previous studies and in line with values reported here,{Costa, 2018 #32} and an attrition rate of 30%, a sample of 210 participants per group (420 in total) would be needed to reject the null hypothesis with probability (power) 0.9 and type I error rate of 5%."

In response to the comment (2) that a recruitment period of over eight years is not feasible, we agree. A future study would need to include additional interventions to improve recruitment by tackling the barriers to recruitment identified here.

In response to comment (3) on details of a potential recruitment intervention, several options are available to reduce the number of exclusions due to surgeon preference. These include an intervention targeted at surgeons to improve their awareness of equipoise and clinical research, an approach that has been described in detail by the QuinteT group (Donovan et al), and reviewed in detail by Townsend et al. References to these are included in the discussion.

In response to comment (4) regarding inclusion of patients with pre-existing arthroplasties, patients with arthroplasties were excluded where their arthroplasty “precluded nail fixation”. Patients with arthroplasties that were compatible with intramedullary nail fixation were included.

In relation to the comment on cost effectiveness of the study (5), feasibility studies necessarily have high per-patient costs as many of the setup and staffing costs are the same as a larger study that recruits more patients. Therefore the per-patient costs of recruiting to a feasibility study do not give an accurate estimate of the potential per-patient costs for a larger study.

In response to comment (6) about examples of other RCTs experiencing similar issues, we have included references to studies by Donovan et al, who summarise challenges across six RCTs. The issues identified by Donovan et al are in line with the issues around surgeon preference and lack of equipoise identified here.

We agree with comment (7), that treatment of intra-articular fractures is very controversial. We included these intra-articular fractures in the study.

We thank reviewer 2 (Mehmet Salih Soylemez) for their comments. In response, details of the patient who was under 50 and their fracture have been added to the paper.

Regards the comment on fragility fracture definition, we note that there is no accepted definition of fragility fracture, and this the study aimed to assess an approach that combined mechanism of injury with age. In the discussion we proposed defining fragility fracture as a fall from standing height, however the appropriateness of this definition will need to be assessed as part of a future study.

We thank reviewer 3 (David Colquhoun) for their comments, and agree it is unfortunate that traditional RCTs are difficult to conduct in this field. We hope that further interventions can improve surgeon participation, and that simplification of recruitment procedures and questionnaires by excluding patients without capacity, and extra training for researchers, can improve recruitment and retention. We agree that a surgeon preference trial is a possibility, as surgeons would only be expected to perform operations they are more confident at. Similar trial designs have been used successfully and unsuccessfully in the past, but are relatively untested.

We thank the reviewer for taking the time to read the statistical analysis plan (SAP), and for their comments on the use of cut-points for continuous variables, potential inclusion of randomisation tests, and the use of a 5% significance level. The SAP for this feasibility study has been written and followed and can no longer be changed, the SAP for any future definite study will take these comments into account.

We also agree that blinding is crucial, and a majority of participants in the study remained blinded. However, due to the unblinding nature of X-rays, medical records and the wide variety of clinical specialities seen during patients’ treatment and recovery covering many months, this can be problematic.

In response to the comment regarding Danone Nutricia we note that Danone Nutricia have not been involved in the current study, or informed of its results.

VERSION 2 – REVIEW

REVIEWER	Hannu T Aro Department of Orthopaedic Surgery and Traumatology, Turku University Hospital and University of Turku, Finland
REVIEW RETURNED	20-Feb-2019

GENERAL COMMENTS

The manuscript is now clear and concise. Without doubt, most traumatologists will agree with the conclusions and appreciate the chance to learn the experience of the investigators.